# Inflammatory Indices and CA 125: A New Approach to Distinguish Ovarian Carcinoma and Borderline Tumors in Suspicious Ovarian Neoplasms from a Retrospective Observational Multicentric Study

**DOI:** 10.3390/medicina61050777

**Published:** 2025-04-22

**Authors:** Carlo Ronsini, Stefano Restaino, Giuseppe Vizzielli, Mariano Catello Di Donna, Giuseppe Cucinella, Maria Cristina Solazzo, Cono Scaffa, Pasquale De Franciscis, Vito Chiantera

**Affiliations:** 1Unit of Gynecologic Oncology, National Cancer Institute, IRCCS, Fondazione “G. Pascale”, 80131 Naples, Italy; mariano.didonna@istitutotumori.na.it (M.C.D.D.); giuseppe.cucinella@istitutotumori.na.it (G.C.); mariacristinasolazzo@gmail.com (M.C.S.); c.scaffa@istitutotumori.na.it (C.S.); vito.chiantera@istitutotumori.na.it (V.C.); 2Unit of Obstetrics and Gynecology, “Santa Maria Della Misericordia” University Hospital, Azienda Sanitaria Universitaria Friuli Centrale, 33100 Udine, Italy; restaino.stefano@gmail.com (S.R.); giuseppevizzielli@yahoo.it (G.V.); 3Unit of Gynaecology and Obstetrics, Department of Woman, Child and General and Specialized Surgery, University of Campania “Luigi Vanvitelli”, 80138 Naples, Italy; pasquale.defranciscis@unicampania.it

**Keywords:** ovarian cancer, borderline ovarian tumor, inflammatory indices, CA 125, diagnostic biomarkers

## Abstract

*Background and Objectives*: This study aimed to evaluate the diagnostic potential of systemic inflammatory indices such as Systemic Inflammation Response Index (SIRI) and Systemic Inflammatory Response (SIR). These were assessed in combination with CA 125 to distinguish ovarian carcinoma (OC) from borderline ovarian tumors (BOT) in patients with suspicious adnexal masses. *Materials and Methods*: A retrospective multicenter observational study including patients undergoing surgery for suspected ovarian neoplasms was conducted. Inclusion criteria required preoperative blood sampling for inflammatory markers and CA 125. SIR-125 and SIRI-125 were developed by combining SIR and SIRI with CA 125 levels. Diagnostic performance was assessed using ROC curve analysis and linear regression models. *Results*: A total of 63 patients (42 BOT, 21 OC) were analyzed. OC patients exhibited significantly higher SIR-125 and SIRI-125 values (*p* < 0.001). ROC analysis demonstrated good diagnostic accuracy, with AUCs of 0.83 (SIR-125) and 0.82 (SIRI-125). SIR-125 showed higher specificity (0.83), while SIRI-125 had superior sensitivity (0.86). *Conclusions*: SIR-125 and SIRI-125 enhance diagnostic differentiation between OC and BOT, providing a simple, cost-effective preoperative tool. Future prospective studies are needed to validate these findings in broader patient populations.

## 1. Introduction

Early diagnosis is one of the most significant limitations related to the clinical management of Ovarian Cancer (OC) [1]. This issue has serious care, prognostic, and quality-of-life consequences, contrasting scenarios in which surgical removal is resolved with lengthy, often unsuccessful courses of treatment [2]. Over the years, many diagnostic models have been proposed to address this issue [3,4,5,6]. One of the most validated and widely used tools is the Assessment of Different NEoplasias in the adneXa (ADNEX) developed by the International Ovarian Tumor Analysis (IOTA) group [7]. This model combines clinical, serologic, and ultrasound data to differentiate the nature of adnexal neoplasms. It has the excellent ability to distinguish between benign and advanced malignant conditions with a sensitivity of >95%. Still, it fails to differentiate benign lesions from OC Stage I, according to FIGO staging [8] and Borderline Ovarian Tumor (BOT). The diagnosis becomes even more complicated when differentiating between these two categories. Faced with a suspected ovarian neoformation, the most difficult differential diagnosis falls precisely between OC and BOT. This diagnosis is not an end in itself. BOT and OC have extremely different survival curves and require different staging treatments [9]. OC diagnosis requires lymph node staging, which is considered a surgical method of high complexity and reserved for specialized centers [10]. Unfortunately, even Frozen Section often fails to meet this diagnostic need, with False Negative rates as high as 30% in some series [11]. Therefore, second-stage interventions are required in OC, to the detriment of health economics and patient health. Therefore, our study aimed to provide additional tools for refining the diagnosis in this scenario using systemic inflammation indices (SIR and SIRI). Many solid neoplasms show an increase in these inflammation indices, which are finding increasing use for diagnostic and prognostic purposes [12,13,14]. As a result, we believe it would be useful to explore the behavior of ovarian carcinoma specifically in the early stages of the disease, highlighting a possible difference in the patient’s inflammatory response and offering an additional possibility for differential diagnosis.

### Objective

To achieve this, we conducted a retrospective observational multicenter cohort study, enrolling patients with suspected adnexal neoplasms who had a concrete risk of non-benign disease and low risk of advanced cancer according to ADNEX, in the absence of suspected extraovarian disease. We then compared the performance of inflammation indices, both alone and combined with the known OC-associated tumor marker, CA 125, between OC and BOT. In this study, we hypothesized that inflammatory indices combined with CA 125 (SIR-125 and SIRI-125) are significantly higher in patients with ovarian cancer than in those with borderline tumors and that these new indices also improve diagnostic ability compared with CA 125 alone. The study also aimed to analyze its diagnostic power by constructing ROC curves and estimating the optimal cut-off to help with the differential diagnosis.

## 2. Materials and Methods

### 2.1. Study Design

We conducted a retrospective observational multicenter cohort study with secondary data from a clinical database, selecting patients with BOT and OC who underwent primary surgery staging at the Gynecologic Oncology unit of the University of Campania Luigi Vanvitelli, Naples, Italy, and the Unit of Obstetrics and Gynecology at “Santa Maria della Misericordia” University Hospital, Udine, Italy. The study used the STROBE statement for observational studies [15]. A dedicated consent form for anonymous data processing and a study-specific consent form were required for all patients treated in the study. According to regulations enforced in the state where the study was conducted, an IRB was unnecessary due to its retrospective nature. The planning, conduct, and reporting of this research are in accordance with the Helsinki Declaration. The study’s primary outcome was to assess the difference in mean inflammatory index (SIR and SIRI) combined with CA 125 between OC and BOT. We evaluated their diagnostic ability by constructing ROC curves. A linear regression model was used to explore the association between the indices and cancer diagnosis.

### 2.2. Setting

In January 2025, we retrospectively collected data from all patients treated for suspected ovarian neoformations in our two institutions between January 2023 and January 2025. All patients had a histological diagnosis of ovarian cancer or borderline ovarian tumors following staging surgery, which included hysterectomy, bilateral adnexectomy, omentectomy, peritoneal biopsy, and, in case of OC diagnosis, lumbo-aortic and pelvic lymphadenectomy. All patients underwent blood sampling 7 days before surgery, with a quantitative white blood cell assay and characterization. All patients underwent a serum CA 125 assay 15 days before surgery. Patients were stratified according to their histological diagnosis and divided into BOT and OC. Two expert anatomopathologists independently confirmed the histological diagnosis. Seventy-one patients met the inclusion criteria. Sixty-three patients were selected according to the required sample size. One patient with ovarian cancer and seven patients with BOT were excluded due to a lack of complete information used as variables.

### 2.3. Participants

Inclusion criteria for participation in the study were: histological diagnosis of OC or BOT, with complete and retrievable anatomopathological information; a complete blood count (CBC) within 7 days before surgery; serum CA 125 dosage within 15 days before surgery; complete surgical staging for the neoplasia; complete information regarding the clinical status at the time of diagnosis, based on a preoperative CT scan or a total body PET scan (performed not more than 30 days before surgery) showing no extra-ovarian localization and reviewed by two independent blinded expert radiologists; and evaluation by the ADNEX model indicating a risk of non-benign cyst ≥ 30% and a risk of disease more advanced than Stage I ≤ 10%.

Patients were excluded if they had any chronic systemic inflammatory condition indicated by a clinical history of chronic inflammatory diseases, including Crohn’s disease, ulcerative colitis, systemic lupus erythematosus, multiple sclerosis, Hashimoto’s thyroiditis, non-alcoholic fatty liver disease, fibromyalgia, chronic kidney disease, hepatitis, osteoarthritis, or psoriasis. Additionally, the exclusion criteria comprised patients with other concurrent or recent (within the last three years) oncological diagnoses, those with disorders leading to excessive corticosteroid production, and individuals who received steroid therapy 30 days preceding blood sample collection. Another exclusion criterion was the concomitant presence of an acute inflammatory state, as evidenced by fever or systemic symptoms 7 days prior to enrollment. Patients with incomplete data and those lacking a definitive histological diagnosis for one of the two targeted conditions were classified under the ‘intention-to-treat’ category. However, no patients met these exclusion criteria.

### 2.4. Variables

The variables examined were Body Mass Index (BMI), as a continuous variable in kg/m^2^; age, a continuous variable expressed in years; side of the adnexal neoformation, identified as left, right, and bilateral; and maximum diameter, a continuous variable expressed in millimeters, evaluated separately by two expert sonographers. The histotype of ovarian neoplasia is understood as a certain histological diagnosis and considered as a binominal variable, divided in two main categories, OC and BOT, and substratified by specific histotype, judged separately by two pathologists. Neutrophils, monocytes, lymphocytes, and platelets were measured as continuous variables, expressed as 10^3^ unit/dL. Serum CA-125 was also considered a continuous variable, expressed in IU/dL. Neutrophils, monocytes, lymphocytes, and platelets were combined into two inflammatory indexes: the Systemic Inflammatory Response (SIR), calculated by multiplying the neutrophil count by the platelet count and dividing by the lymphocyte count; and the Systemic Inflammatory Response Index (SIRI), calculated by multiplying the monocyte count by the platelet count and dividing by the lymphocyte count. Both indices were expressed as continuous variables. Finally, two new indices were created by multiplying SIR by CA 125 and normalizing it by dividing the result by the correction factor 1000 (SIR-125) and multiplying SIRI by CA 125 (SIRI-125). These two indices were also used as continuous variables.

### 2.5. Laboratory

Peripheral blood samples (3.0 mL) were collected from the ulnar vein within seven days before surgery for hematological analysis. These samples were drawn using a sterile vacuum collection system and immediately placed in tubes containing ethylenediaminetetraacetic acid (EDTA) to prevent coagulation. The samples were gently inverted several times to ensure proper anticoagulation and stored at 4 °C until processing. Within two hours of collection, they were analyzed using an automated hematology analyzer to determine total neutrophils, lymphocytes, monocytes, eosinophils, basophils, and platelet counts, with results expressed as 10^3^ units/dL.

Additionally, serum CA 125 levels were measured from blood samples collected within 15 days before surgery. These samples were drawn into serum separator tubes (SST), left to clot at room temperature for 30 min, and then centrifuged at 3000× *g* for 10 min to separate the serum. The serum was then aliquoted and stored at −80 °C until analysis. Quantification was performed using an electrochemiluminescence immunoassay (ECLIA) on a Cobas e411 analyzer (Roche Diagnostics, Basel, Switzerland), ensuring high sensitivity and specificity in detecting CA 125 levels.

All sample processing and analyses were conducted in-house in the two participating facilities, following standardized laboratory protocols to ensure accuracy and reproducibility.

### 2.6. Statistical Analysis

The distribution of continuous variables was assessed using the Kolmogorov–Smirnov test to evaluate normality. Nominal variables were presented as absolute frequencies and percentages, and comparisons between groups were performed using Fisher’s exact and Chi-square tests. Continuous variables were summarized as medians with interquartile ranges and analyzed using the Wilcoxon test for two-group comparisons. The Kruskal-Wallis test was applied when comparing more than two independent groups. Patients were classified into OC and BOT based on their histological diagnosis. The study’s null hypothesis assumed no significant difference in mean SIR-125 and SIRI-125 values between OC and BOT patients (H_0_: µ_1_ = µ_2_; H_1_: µ_1_ ≠ µ_2_, two-sided tests). The sample size was calculated to detect a minimum difference of 0.76 standard deviations in the mean values of SIR-125 and SIRI-125 between OC and BOT groups. Since no preliminary data were available, this estimate was based on the assumption of a marked difference between the two groups. To ensure 80% statistical power and a significance level of α = 0.05, the calculation determined a minimum sample of 21 OC patients. A 2:1 sampling ratio was chosen to improve the stability of the estimates, enrolling 42 BOT patients for a total of 63 patients. The calculation was performed using R (pwr package). We conducted a multivariate logit regression to demonstrate a correlation between the parameters examined and alterations in inflammation indices regression. The model’s significance was assessed using the maximum likelihood method. ROC curve analysis was conducted to evaluate the diagnostic performance of SIR-125 and SIRI-125 in distinguishing histological subtypes. The area under the curve (AUC) was computed to assess overall diagnostic accuracy, employing the 2000 bootstrap resampling method. The Youden Index was determined to identify the optimal cutoff value for SIR-125. Boxplots were used to visualize the distribution of continuous variables for the respective outcome groups. All statistical analyses were conducted using R software and RStudio (version 2024.12.0 + 467). ROC curves and AUC values were generated using the ROC package, while the Youden Index was applied to establish the best cutoff points. A *p*-value < 0.05 was considered statistically significant. The anonymized dataset used in this study is provided in the Appendix A.

### 2.7. Risk of Bias

Multivariate regression studies were conducted with a combination of all variables present to minimize confounders. The individual models obtained were compared using adjusted R2 and Bayesian Information Criteria (BIC). The best model was chosen based on the lowest expressed value of BIC. Data analysis was first conducted by CR and then reviewed in a blind manner by MCDD, who was unaware of the study’s objective. No missing data were present in the outcomes of interest.

## 3. Results

In accordance with the calculated sample power, we retrospectively enrolled 63 patients meeting the inclusion criteria, 42 BOT and 21 OC, based on a pre-established proportionality of 2:1. No statistically significant difference was found between the two groups regarding age, BMI, side and diameter of lesion, and menopausal status. The characteristics of the population are summarized in Table 1. The group with ovarian cancer showed higher mean values of CA 125, neutrophils, monocytes, and platelets. The serum analysis levels of the population are summarized in Table 1.

### 3.1. Outcomes

The primary outcome was to estimate the difference in mean value of SIR-125 and SIRI-125 between BOT and OC. Previously, an evaluation of SIR and SIRI was performed. SIR and SIRI were higher in the OC group (SIR 1347 vs. 699, *p* = 0.012; SIRI 2.73 vs. 1.36 *p* = 0.04). Figure 1 presents the values of SIR, SIRI, and CA 125 in a boxplot.

Then, SIR-125 was obtained by multiplying SIR by the value of CA 125. The value was divided by an arbitrary correction value of 1000 to improve legibility. SIR-125 was statistically significantly higher in OC patients (92 vs. 14, *p* < 0.001). SIRI-125 was constructed by multiplying SIRI by CA 125 values. This second index was also statistically significantly higher among OC (158 vs. 24, *p* < 0.001). These results are summarized in Table 2 and presented in Figure 2.

### 3.2. Logit Regression

A logit regression model was constructed to evaluate the association between ovarian cancer and the newly developed inflammation indexes, with SIR-125 and SIRI-125 included as independent variables. The analysis revealed a statistically significant association between OC and both inflammatory markers. Specifically, both SIR-125 and SIRI-125 demonstrated an odds ratio of 1.001 with a 95% confidence interval of 1.001–1.001 (*p* = 0.002 for SIR-125; *p* = 0.006 for SIRI-125). These results are shown in Table 3.

### 3.3. ROC Curve

The differential diagnostic ability between BOT and OC of the two created indices was evaluated by constructing two independent ROC curves, as shown in Figure 3.

The AUC related to OC diagnosis was 0.83 (95% CI: 0.70–0.94) for the SIR-125 index, and 0.82 (95% CI: 0.68–0.93) for the SIRI-125 index. Youden’s method was used to calculate the best diagnostic cut-off for SIR-125 and SIRI-125 at 37.67 (rounded to 38) and 61.41 (rounded to 61), respectively. SIR-125 showed a higher specificity (0.83 vs. 0.79) while SIRI-125 showed a higher sensitivity (0.86 vs. 0.76). All data about cut-off performance are reported in Table 4 and Table 5.

## 4. Discussion

### 4.1. Interpretation of Results

Our study showed that on overage, inflammation indices are higher in OC patients than in BOT patients. This result is due to an increase in neutrophils, monocytes, and platelets, while the value of lymphocytes remains superimposable. The combination of these white blood cell populations forms the basis for the common inflammation indices, SIR and SIRI, and explains why they are higher. Not surprisingly, CA 125 also increased, consistent with what is widely known in the scientific literature. Innovation in our research arises from the combination of these factors. Not surprisingly, combining the SIR and SIRI index with the value of CA 125 strengthens this evidence, showing a strong correlation to OC in regression models. Finally, these two new indices, SIR-125 and SIRI-125, show good diagnostic performance in the differential diagnosis of ovarian carcinoma and OC in suspicious adnexal neoformations. The biological reasons behind these results rest on several hypotheses. Firstly, the tumor microenvironment shows pro-inflammatory features, which promote tumor progression [16]. Moreover, tumor metabolism, which is based on mechanisms of hypoxia and oxidative stress, may also reverberate in the subject’s immune response [17]. Interestingly, our set of patients was limited to ovarian-confined disease. This limitation makes it more likely that the observed inflammatory response is due to the tumor’s nature, eliminating any potential confounding from metastatic dissemination. Thus, intrinsic mechanisms of carcinomatous degeneration determine different immune responses in OC and BOT. Finally, it should be noted that the sample showed no statistically significant differences in lesion bilaterality or maximum size. These two factors, hypothetically, could have affected the actual tumor volume and its ability to trigger an immune response.

### 4.2. Comparison with Existing Literature

Various solid tumors are known to trigger an inflammatory response from the host organism [18]. These indices of systemic inflammation are already elevated at the origin of tumor progression. Our group recently demonstrated how in endometrial carcinoma, endometrial carcinoma has a similar attitude toward atypical endometrial hyperplasia as that shown by OC for BOT in this study [19]. Moreover, it is also known that local tumor infiltration may be the first trigger for the immune system [20]. Our results are in line with these principles. However, it is interesting to note that OC appears to go through different phases in modulating the immune response. As recently reported by Blanc-Durand et al., in the early stages of the disease, OC elicits an immune response mediated by Natural Killer (NK) cells and tumor-infiltrate T lymphocytes. Subsequently, the chronification of the inflammatory response promotes neovascularization, metastasization, and immunoevasion [21]. Such evidence is also supported by research on extracellular vesicles in OC, which has shown an overproduction of IL-6 and TGF β1 in the early stages of ovarian carcinoma [22,23]. Moreover, a recent prospective study by Santiago et al. showed that ovarian carcinomas have a higher production of pro-inflammatory cytokines than patients without disease [24]. Finally, several papers have also tested the prognostic value of indices of systemic inflammation in OC, showing a correlation with both overall survival and platinum-free survival [25]. This evidence is focused on the same subject matter as our study, but from a different perspective. Our work focuses on different distributions of these indices to exploit their possible diagnostic power.

### 4.3. Clinical Implication

Our two new indices were created to help in the differential diagnosis of BOT and OC in patients with suspicious adnexal neoformations. It should be noted that this particular patient setting presents a significant challenge for clinicians. In fact, several models propose methods that reliably predict the presence of an OC in an adnexal formation [3,4]. Most of these models are based on a combination of imaging data (ultrasound and MRI) and CA 125 [5,6]. The performance of these tools has improved over the years, reaching ROCs with AUCs exceeding 90%. However, the great potential of their diagnostic power is mainly exercised in distinguishing benign adnexal formations from advanced carcinomas. Performance is already reduced when asked to differentiate between benign formations and neoplasms confined to the ovary. It tapers down even more when asked to differentiate between BOT and OC. This finding is already evident if one considers the ADNEX model, widely used in clinical practice, which is based on a combination of ultrasound and CA 125 features. It provides results in the form of diagnostic probability, losing much of its accuracy when attempting to differentiate between BOT and OC [7]. It should be noted that this differential diagnosis suffers the most from false positives and false negatives in the Frozen Section, putting patients at high risk of overtreatment or stadiative second-look procedures [11]. Therefore, this category of patients, the subject of our study, would benefit from diagnostic tools that facilitate this distinction. In this scenario, assessing inflammatory status could help judge cases at high risk of ovarian cancer. In fact, from a healthcare organization perspective, the timing of intervention and the modalities and centers to treat it change drastically between BOT and OC. Our indices are a convenient and replicable tool that can assist with this diagnosis. They are based on routine preoperative investigations that do not require any technological sophistication, making this tool widely applicable. To optimize its usability, we also made a mathematical correction to SIR-125 to algebraically decrease values by dividing the result by 1000. It should be noted that the present authors do not propose to replace this evidence with the well-established use of previously mentioned diagnostic tools. Still, they can provide additional help in refining the diagnosis. Moreover, the SIR-125 and SIRI-125 indices appear to complement each other. The former exhibits high specificity and negative predictive value, making it particularly useful in confirming diagnostic suspicions and limiting the risk of stadiative surgical overtreatment, such as lumbo-aortic lymphadenectomy. By contrast, SIRI-125 shows high sensitivity and positive predictive value, making it a good tool for early diagnosis.

### 4.4. Strengths and Limitations

Our study finds its strength in the strong statistical significance of the results, methodological rigor, and easy large-scale reproducibility. However, it has limitations inherent to the study’s retrospective nature, which does not completely eliminate confounding factors potentially associated with variations in the immune system and inflammatory response. Despite strict exclusion criteria, the sample could be influenced by environmental factors specific to the geographic region where the study was conducted (Italy), which may interact with the immune system [26]. Another major limitation is undoubtedly related to the absence of a disease-free control arm. Although the purpose of the study was to highlight differences between BOT and OC due to current diagnostic difficulties, extending it to a healthy population would further its validate clinical usability. However, in our opinion, our research should serve as a springboard for conducting larger prospective studies, including additional parameters for inflammation such as lymphocyte population type or C-reactive protein.

## 5. Conclusions

In conclusion, our study showed that systemic inflammatory indices combined with CA 125, especially SIR-125 and SIRI-125, can improve diagnostic ability in distinguishing between OC and BOT in suspected adnexal neoformation. These new parameters showed good diagnostic accuracy, with SIR-125 having high specificity and SIRI-125 having high sensitivity. The study focused on a patient population already screened by the ADNEX model. Future prospective studies involving patients without suspected disease will be needed to improve the usability of these results.

## Figures and Tables

**Figure 1 medicina-61-00777-f001:**
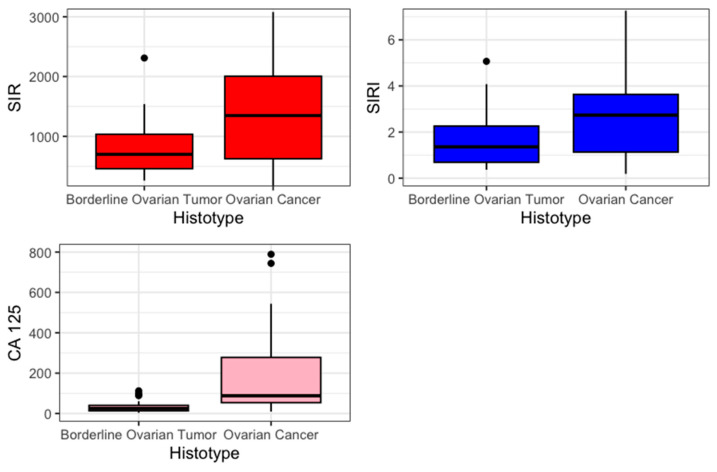
Boxplot.

**Figure 2 medicina-61-00777-f002:**
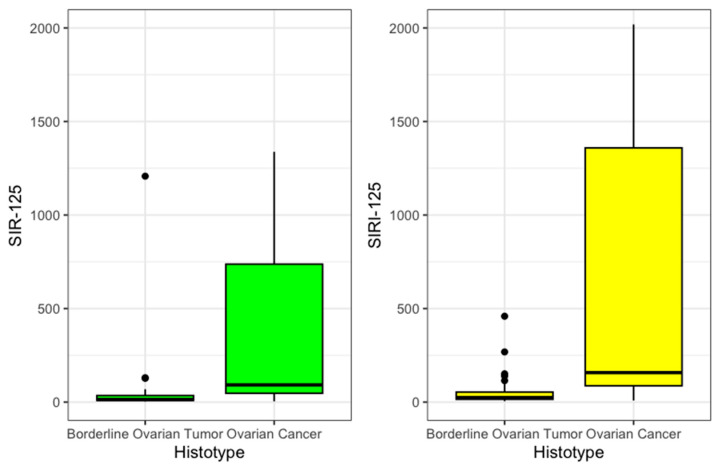
SIR125 and SIRI 125 boxplot.

**Figure 3 medicina-61-00777-f003:**
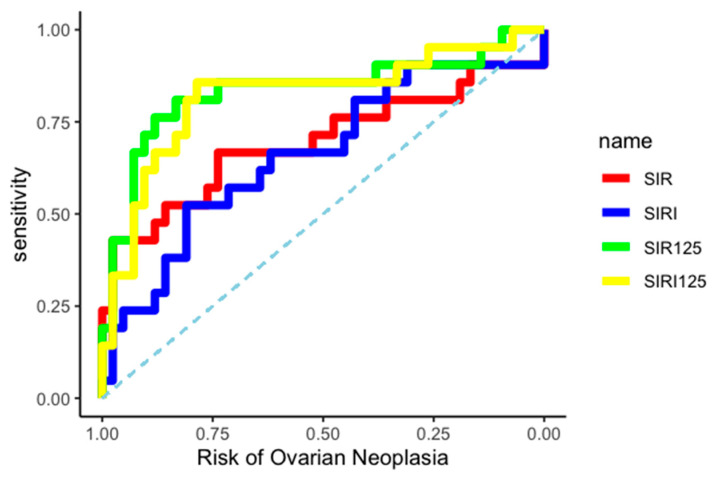
ROC curves.

**Table 1 medicina-61-00777-t001:** Patients’ characteristics and serum analytes.

Characteristic	Borderline Ovarian Tumor, N = 42 ^1^	Ovarian Cancer, N = 21 ^1^	*p*-Value ^2^
Age	51 (38, 64)	56 (49, 63)	0.137
BMI	23.0 (19.4, 27.6)	25.7 (23.0, 29.3)	0.078
Missing	2	0	
Menopouse	20 (48%)	14 (67%)	0.187
Side			0.067
Left	10 (24%)	10 (48%)	
Right	22 (52%)	10 (48%)	
Bilateral	10 (24%)	1 (4.8%)	
Maximum Diameter	85 (44, 150)	108 (69, 131)	0.570
Histotype			
Endometrioid BOT	1 (2.4%)	-	
Serous BOT	22 (52%)	-	
Mucinous BOT	19 (45%)	-	
HGSOC	-	9 (43%)	
LGSOC	-	4 (19%)	
Clear Cell	-	3 (14%)	
Endometrioid	-	1 (4.8%)	
Mucinous Expansive	-	2 (9.5%)	
Stromal	-	1 (4.8%)	
Serum CA-125	25 (13, 40)	88 (54, 278)	**<0.001**
Neutrophils (10^3^ units/dL)	4.5 (3.5, 6.2)	6.6 (4.6, 7.2)	**0.034**
Lymphocytis (10^3^ units/dL)	1.68 (1.24, 2.07)	1.66 (1.28, 2.16)	0.919
Monocytis (10^3^ units/dL)	0.44 (0.35, 0.59)	0.60 (0.52, 0.70)	**0.046**
Platelets (10^3^ units/dL)	234 (199, 276)	300 (262, 371)	**0.003**

^1^ Median (Q1–Q3); n (%). ^2^ Wilcoxon rank sum test; Fisher’s exact test. Bolded numbers indicate statistically significant values (*p* < 0.05).

**Table 2 medicina-61-00777-t002:** Outcomes.

Characteristic	Borderline Ovarian Tumor, N = 42 ^1^	Ovarian Cancer, N = 21 ^1^	*p*-Value ^2^
SIR-125	14 (9, 35)	92 (47, 737)	**<0.001**
SIRI-125	24 (15, 54)	158 (87, 1359)	**<0.001**
SIR	699 (460, 1034)	1347 (626, 2004)	**0.012**
SIRI	1.36 (0.69, 2.26)	2.73 (1.13, 3.63)	**0.040**

^1^ Median (Q1–Q3). ^2^ Wilcoxon rank sum exact test. SIR-125: Multiplication of SIR by the value of CA 125/1000. SIRI-125: Multiplication of SIRI by the value of CA 125.

**Table 3 medicina-61-00777-t003:** Logit regression for ovarian cancer.

Characteristic	Estimate	Standard Error	t-Value	ODDS	95% CI	*p*-Value
SIR-125	2.124	6.46	3.288	1.001	1.001–1.001	**0.002**
SIRI-125	9.625	3.371	2.855	1.001	1.001–1.001	**0.006**

**Table 4 medicina-61-00777-t004:** SIR-125 performance.

SIR-125 ^1^	Estimated	Lower Limit (95%)	Upper Limit (95%)
Sensitivity	0.76	0.53	0.92
Specificity	0.83	0.69	0.93
PPV	0.70	0.47	0.87
NPV	0.88	0.73	0.96
Odds Ratio	16	4.4	58.19
Accuracy	0.81	0.69	0.89

PPV: Positive Predictive Value. NPV: Negative Predictive Value. ^1^ 38, Calculated by Youden Index.

**Table 5 medicina-61-00777-t005:** SIRI-125 performance.

SIR-125 ^1^	Estimated	Lower Limit (95%)	Upper Limit (95%)
Sensitivity	0.86	0.64	0.97
Specificity	0.79	0.63	0.90
PPV	0.67	0.46	0.84
NPV	0.92	0.78	0.98
Odds Ratio	22	5.28	91.68
Accuracy	0.81	0.69	0.90

PPV: Positive Predictive Value. NPV: Negative Predictive Value. ^1^ 61, Calculated by Youden Index.

## Data Availability

All data and the methodological process for their calculation can be supplied upon explicit request to the corresponding author and are available as an ‘.R’ file uploaded in the Appendix A of this manuscript.

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
