# Peer review of "Inflammatory Indices and CA 125: A New Approach to Distinguish Ovarian Carcinoma and Borderline Tumors in Suspicious Ovarian Neoplasms from a Retrospective Observational Multicentric Study"

_medicina, 2025, doi:10.3390/medicina61050777_

Round 1
Reviewer 1 Report
Comments and Suggestions for Authors
The manuscript entitled “Inflammatory Indices and CA 125: A New Approach to Distinguish Ovarian Carcinoma and Borderline Tumors in Suspicious Ovarian Neoplasms from a Retrospective Observational Multicentric Study” is well structured and effectively written.
The study examines the to evaluate the diagnostic potential of systemic inflammatory indices (SIRI, SIR) combined with CA125 in distinguishing ovarian carcinomas (OC) from borderline ovarian tumors (BOT) in patients.
A total of 63 patients (42 BOT, 21 OC) were analyzed in the study. These studies are SIR-125 and SIRI-125 enhance the diagnostic differentiation between OC and BOT, providing a simple, cost-effective preoperative tool.
This manuscript is suitable for acceptance pending minor revisions to enhance clarity and address specific feedback:
- How SIR-125 and SIRI-125 compare with other established biomarkers or imaging techniques in differentiating OC from BOT?
- What are the clinical implications of using SIR-125 and SIRI-125 in routine practice. Could the authors discuss these aspects in the context of finding?
- Could this study have a combining additional inflammatory or tumor markers further improve diagnostic accuracy.
- How do these indices perform across different histological subtypes of OC and BOT.
- This study only considers CA-125, SIR and SIRI, while other potential inflammatory or tumor markers are not evaluated?
- Several sentences could be rephrased for better clarity and readability.
Line 67: “To do so” is informal. Use some other phrase like “To achieve this”
Line 68: Neoformations is an uncommon word. Neoplasm would be more precise in medical term.
Line 183: Sentence in objective section is too long and should be split into two or restructured for better readability.
Line 81: “A clinical database” is grammatically correct.
Line 82: Primary surgical staining.
Line 85: Dedicated consent forms for anonymous data processing and study-specific consent were required.
Line 109: “having undergone a complete blood count (CBC) test within 7 days before surgery”.
Author Response
Dear Reviewer,
Thank You for taking the time to review our manuscript and for your comments. They are crucial and valuable to us in raising the quality standard of our work.
We wanted to inform You that we have made a general revision of the English and grammar. In addition, a specification for Your revisions is below:
1) How SIR-125 and SIRI-125 compare with other established biomarkers or imaging techniques in differentiating OC from BOT?
1) The aim of our study was to explore the inflammatory response in Borderline and ovarian cancer, and it represents an initial study that can be the basis for future investigations. We are currently conducting a similar study with assessment of the nutritional status of patients as well. However, to date, due to the novelty represented by the study, there are no correlations described in the literature between our research object and what is already known
2) What are the clinical implications of using SIR-125 and SIRI-125 in routine practice. Could the authors discuss these aspects in the context of finding?
2) In our opinion, adding the assessment of inflammatory status in cases of ovarian neoplasm of a nature to be determined can help weigh the risk and set priorities for intervention, especially in centers with high volume of care, where the lack of an accurate tool for differential diagnosis between BOT and Carcinoma, may condition clinical conduct. We have better contextualized this concept in the “Discussion” (lines 328-330).
3) Could this study have a combining additional inflammatory or tumor markers further improve diagnostic accuracy.
3) This study represents a first series and analysis of the most common inflammatory factors available to the clinician. The construction of prospective models with confounding limitation and evaluation of additional markers of inflammation (lymphocyte type, PCR, etc.) may represent new research frontiers. We added this concept to the limitations of the study (lines 357-358).
4) How do these indices perform across different histological subtypes of OC and BOT.
4) Unfortunately, the low numbers of rare histotypes made no statistical investigation possible. Future studies focusing on a single lesion type will be needed to clarify these doubts.
5) This study only considers CA-125, SIR and SIRI, while other potential inflammatory or tumor markers are not evaluated?
5) Because of the retrospective nature of the study, parameters most easily obtained in clinical routine were considered. Undoubtedly, future prospective studies should include additional factors of inflammation
6) Several sentences could be rephrased for better clarity and readability.
6) a general revision of English was conducted pr improve legibility. Thanks of the proposed potential lexical improvements. The following have been accepted and modified in the attached version of the manuscript
The rewritten and corrected version of the manuscript is also included in the attached file. We highlighted any changes made.
Thank you very much for your advice and comments. We hope we have complied with your requests.
Reviewer 2 Report
Comments and Suggestions for Authors
- It is required to show full names of SIR and SIRI and their calculation in the abstract section. What is the reason for adopting lymphocytes for constituting the SIR and SIRI, considering that lymphocytes have no value in differentiating BOT and OC? Are SIR, SIRI, SIR-CA125, and SIRI-CA125 normally distributed? How about the collinearity between SIR-CA125 and SIRI-CA125?
- Authors selected 42 patients with BOT and 21 patients with OC out of the database. However, the selection method from the initial database is not available.
- “Table 125” on page 7 (line 227) likely refers to SIR-CA125. In addition, please confirm whether lymphocytis, monocytis, and platets are correct.
- In Table 3, a linear regression model was constructed to evaluate the relationship between OC and the inflammation indexes. I am not sure whether linear regression model can be applied to this analysis, as the d데둥둣 variable (diagnosis) is categorical.
- In terms of clinical implication, it is recommended to compare the ADNEX model with inflammatory indexes (combination of SIR-CA125 and SIRI-CA125). In addition, it is also recommended to compare the ADNEX model with combination of ADNEX model alongside inflammatory indexes.
The English could be improved to more clearly express the research.
Author Response
Dear Reviewer,
Thank You for taking the time to review our manuscript and for your comments. They are crucial and valuable to us in raising the quality standard of our work.
We wanted to inform You that we have made a general revision of the English and grammar. In addition, a specification for Your revisions is below:
- It is required to show full names of SIR and SIRI and their calculation in the abstract section. What is the reason for adopting lymphocytes for constituting the SIR and SIRI, considering that lymphocytes have no value in differentiating BOT and OC? Are SIR, SIRI, SIR-CA125, and SIRI-CA125 normally distributed? How about the collinearity between SIR-CA125 and SIRI-CA125?
- SIR and SIRI were expressed explicitly in the abstract. The use of lymphocytes was necessary because SIR and SIRI are established inflammatory models used in a variety of aspects of clinical practice, including oncology. Therefore, changing the construction of these parameters would have been a bias in our opinion. Moreover, the absence of a different distribution of lymphocytes in the two populations does not negate the fact that variations in SIR and SIRI may be related to fluctuations in other parameters that go into their construction, justifying the different results obtained. The distribution of SIR and SIRI is not normal, and so the statistical tests used were not parametric (We have reported in detail the tests for evaluating the distribution and the tests used in Section 2.6). Sharing the same denominator there is colinearity between SIR and SIRI, ontologically. However, we tested the diagnostic power of both as this study is a first assessment of their diagnostic power.
Authors selected 42 patients with BOT and 21 patients with OC out of the database. However, the selection method from the initial database is not available.
2. Thanks. 71 hypothetical patients met the inclusion criteria. 8 were excluded in accordance with the required numerosity of 63, eliminating patients with less complete data. We added this information in section 2.2
3. Table 125” on page 7 (line 227) likely refers to SIR-CA125. In addition, please confirm whether lymphocytis, monocytis, and platets are correct.
- It refers to the value of CA 125, as shown in the table. The remaining data are correct.
4. In Table 3, a linear regression model was constructed to evaluate the relationship between OC and the inflammation indexes. I am not sure whether linear regression model can be applied to this analysis, as the d데둥둣 variable (diagnosis) is categorical.
4. We agree that the correct model is the Logit regression model. The results were calculated with this model. We have corrected the misprint.
5. In terms of clinical implication, it is recommended to compare the ADNEX model with inflammatory indexes (combination of SIR-CA125 and SIRI-CA125). In addition, it is also recommended to compare the ADNEX model with combination of ADNEX model alongside inflammatory indexes.
5. We agree with this observation. However, a direct comparison between the ADNEX model and inflammatory indices is not possible in this series of patients for 2 reasons. 1, There is a lack of data on adnex calculation in numerous patients. 2. Our model uses a diagnostic cut-off derived from ROC curves, whereas the ADNEX model is expressed in risk probabilities. Undoubtedly, future prospective studies can perform a direct comparison. This has been reported in the Discussion section.
The rewritten and corrected version of the manuscript is also included in the attached file. We highlighted any changes made.
Thank you very much for your advice and comments. We hope we have complied with your requests.
Reviewer 3 Report
Comments and Suggestions for Authors
Dear authors and editor,
Firstly I would like to thank you for the opportunity of heaving read this manuscript as this matter is of great interest.
Secondly, I have some some questions about this mansucript.
Line 118 - Have you considered acute inflammation as a exclusion factor?
Line 118-128 I consider that the paragraph in unnecessary as you did not have any patients to exclude.
Line 139 - please rephrase
Line 151-158 why is it important?
Table 1 should be split in more tables for easier understanding
Kind regards,
Author Response
Dear Reviewer,
Thank You for taking the time to review our manuscript and for your comments. They are crucial and valuable to us in raising the quality standard of our work.
We wanted to inform You that we have made a general revision of the English and grammar. In addition, a specification for Your revisions is below:
1. Line 118 - Have you considered acute inflammation as a exclusion factor?
1. Yes , acute inflammation status was an exclusion criteria, as stated in Methods
2. Line 118-128 I consider that the paragraph in unnecessary as you did not have any patients to exclude.
2. Thank you, but we supplemented with intention-to-treat patients because of 71 eligible patients, 63 were enrolled to meet the calculated sample size. Therefore, we retained the paragraph and specified how the patients were selected
3. Line 139 - please rephrase
3. We made a general correction of English to improve readability
4. Line 151-158 why is it important?
4. In our judgment, also specifying the mode of assay of the variables taken into analysis is respectful to the reader and increases the rigor of the exposition to optimize reproducibility even in less developed clinical settings. However, if in your judgment this information is unnecessary, we can remove it.
5. Table 1 should be split in more tables for easier understanding
5. Thank You. We created a Table 1.1 dedicated to serum analysis
The rewritten and corrected version of the manuscript is also included in the attached file. We highlighted any changes made.
Thank you very much for your advice and comments. We hope we have complied with your requests.
Round 2
Reviewer 2 Report
Comments and Suggestions for Authors
- The spelling of the variable names in Table 1.1 should be corrected.
- The paragraph under the “3.2 Logit regression” requires revision to improve readability. In addition, the foot note of Table 3 should be removed.
Thank you.
Comments on the Quality of English LanguageThe English could be improved to more clearly express the research.
Author Response
Dear Reviewer,
Thank You for taking the time to review our manuscript and for your comments. They are crucial and valuable to us in raising the quality standard of our work.
We wanted to inform You that we have made a general revision of the English and grammar. In addition, a specification for Your revisions is below:
Q: The spelling of the variable names in Table 1.1 should be corrected.
A: Thank You, we changed them.
Q: The paragraph under the “3.2 Logit regression” requires revision to improve readability. In addition, the foot note of Table 3 should be removed.
A: Thank you. We removed the footprint and rearranged the paragraph to improve the readability.
The rewritten and corrected version of the manuscript is also included in the attached file. We highlighted any changes made.
Thank you very much for your advice and comments. We hope we have complied with your requests.
Reviewer 3 Report
Comments and Suggestions for Authors
Dear authors,
I would like to thank you for the revised version. I believe that the manuscript should be accepted in the revised form.
Kind regards,